# Spatial and Quantitative Analysis of Tumor-Associated Macrophages: Intratumoral CD163-/PD-L1+ TAMs as a Marker of Favorable Clinical Outcomes in Triple-Negative Breast Cancer

**DOI:** 10.3390/ijms232113235

**Published:** 2022-10-31

**Authors:** Hajime Shinohara, Maki Kobayashi, Kumiko Hayashi, Daichi Nogawa, Ayaka Asakawa, Yae Ohata, Kazuishi Kubota, Hisashi Takahashi, Miyuki Yamada, Masanori Tokunaga, Yusuke Kinugasa, Goshi Oda, Tsuyoshi Nakagawa, Iichiroh Onishi, Yuko Kinowaki, Morito Kurata, Kenichi Ohashi, Masanobu Kitagawa, Kouhei Yamamoto

**Affiliations:** 1Department of Gasrointestinal Surgery, Graduate School of Medicine and Dentistry, Tokyo Medical and Dental University, 1-5-45 Yushima, Bunkyo-ku, Tokyo 113-8519, Japan; 2Molecular Pathology Group, Translational Research Department, Daiichisankyo RD Novare, 1-16-13 Kitakasai, Edogawa-ku, Tokyo 134-0081, Japan; 3Department of Specialized Surgery, Graduate School of Medicine and Dentistry, Tokyo Medical and Dental University, 1-5-45 Yushima, Bunkyo-ku, Tokyo 113-8519, Japan; 4Department of Comprehensive Pathology, Graduate School of Medicine and Dentistry, Tokyo Medical and Dental University, 1-5-45 Yushima, Bunkyo-ku, Tokyo 113-8519, Japan; 5Department of Thoracic Surgery, Graduate School of Medicine and Dentistry, Tokyo Medical and Dental University, 1-5-45 Yushima, Bunkyo-ku, Tokyo 113-8519, Japan; 6Science for Life Laboratory, Department of Medical Biochemistry and Microbiology, Uppsala University, 75236 Uppsala, Sweden; 7Translational Science Department, Daiichi Sankyo, Inc., Basking Ridge, NJ 07920, USA; 8Department of Human Pathology, Graduate School of Medicine and Dentistry, Tokyo Medical and Dental University, 1-5-45 Yushima, Bunkyo-ku, Tokyo 113-8519, Japan

**Keywords:** breast cancer, tumor microenvironment, tumor-associated macrophage, multiplex immunohistochemistry, CD68, CD163, PD-L1

## Abstract

Tumor-associated macrophages (TAMs) and abnormalities in cancer cells affect cancer progression and response to therapy. TAMs are a major component of the tumor microenvironment (TME) in breast cancer, with their invasion affecting clinical outcomes. Programmed death-ligand 1 (PD-L1), a target of immune checkpoint inhibitors, acts as a suppressive signal for the surrounding immune system; however, its expression and effect on TAMs and the clinical outcome in breast cancer are unknown. In this study, we used high-throughput multiple immunohistochemistry to spatially and quantitatively analyze TAMs. We subjected 81 breast cancer specimens to immunostaining for CD68, CD163, PD-1, PD-L1, CD20, and pan-CK. In both stromal and intratumoral areas, the triple-negative subtype had significantly more CD68/CD163, CD68/PD-L1, and CD163/PD-L1 double-positive cells than the estrogen receptor (ER)/progesterone receptor (PR) subtype. Interestingly, a higher number of CD68+/PD-L1+/CK-/CD163- TAMs in the intratumoral area was correlated with a favorable recurrence rate (*p* = 0.048). These findings indicated that the specific subpopulation and localization of TAMs in the TME affect clinical outcomes in breast cancer.

## 1. Introduction

Breast cancer is one of the most prevalent malignancies worldwide. Surgery, chemotherapy, hormone therapy, and human epidermal growth factor receptor 2 (HER2)-targeted immunotherapy are common treatments for breast cancer [1,2]. In addition to traditional treatments, cancer immunotherapy, including immune checkpoint inhibitors, is anticipated to increase patient survival [3,4,5,6]. However, because of metastasis and recurrence, the mortality rate of patients with breast cancer remains significant. Therefore, it is crucial to understand the molecular and microenvironmental aspects of tumors in order to develop novel therapeutic strategies.

Several different cell types exist in the tumor microenvironment (TME), including lymphocytes, macrophages, fibroblasts, endothelial cells, and pericytes [7]. Tumor development is known to be directly correlated with an intricate network of various cells and signaling pathways [7]. Macrophages in the tumor microenvironment are known as “tumor-associated macrophages (TAMs)” [8,9]. Macrophages are the most common infiltrating leukocytes in breast cancer, accounting for 50% of cells in the TME in breast cancer [10,11]. Indeed, TAMs have been shown to play an important role in tumor progression by expressing various effector molecules that inhibit the antitumor immune response [8]. According to their function, TAMs are broadly classified into M1 and M2 macrophages; M2 macrophages share many features with alternative anti-inflammatory macrophages [12], and CD163 is often used as their specific marker [13,14].

Several studies have explored the role of TAMs and their clinical significance in breast cancer, and most have indicated that TAM invasion is an unfavorable prognostic factor [14,15,16,17,18]. As TAMs have been implicated in tumor progression in breast cancer, they have attracted increased attention as potential therapeutic targets. Breast cancer TAMs inhibit succinate dehydrogenase in breast cancer cells, promoting carcinogenesis [19]. In addition, TAMs have been shown to promote immunosuppression in laryngeal squamous cell carcinoma [20] and suppress NK cell activation [21].

With respect to programmed death-ligand 1 (PD-L1), the target axis molecule of immune checkpoint inhibitor (ICI), TAMs have been reported to promote the expression of PD-L1 in pancreatic cancer [22]. Furthermore, PD-L1 antibodies disrupted STAT3/NFκB signaling in M2 macrophages [23], which were also reported to alter T-cell metabolism and induce apoptosis via the PD-L1/PD-1 axis [24]. Moreover, the expression of PD-L1 in TAMs has been reported as a potential target molecule for ICI in liver cancer [25].

Based on these findings, the expression of PD-L1 by TAMs might also affect the TME and be a potential therapeutic target in breast cancer. However, neither TAMs nor the expression of PD-L1 have been quantified, and their clinicopathological significance in breast cancer have been studied in detail. One reason for this is a technical problem. The single-marker staining technique does not allow evaluation in a single section, which raises issues regarding reproducibility and accuracy of the results. In addition, the fact that quantitative evaluation is not possible with conventional techniques has also hindered precise analysis. To solve this problem, high-throughput immunohistochemical analysis (HTIA), which combines multicolored immunohistochemistry (IHC) and quantitative analysis, has been developed [26,27]. HTIA has been reported in various cancers for tumor-infiltrating lymphocytes (TILs), which are important components of the TME [27,28,29,30]. Despite this, there is only one detailed study of TAMs using HTIA across all cancers [31]. In this study, we performed a detailed HTIA of TAM profiles in breast cancer and analyzed the relationship between TAM subpopulations and clinical outcomes by subtype.

## 2. Results

### 2.1. Clinicopathological Features

The study included 81 patients diagnosed with breast invasive ductal carcinoma: 41 with T stage I, 31 with T stage II, 7 with T stage III, and 2 with T stage IV. All patients were female, and the median age was 61 years (range 33–88 years). Lymph node metastasis was positive in 23 patients, negative in 50 patients, and unknown in 8 patients. Immunostaining showed the estrogen receptor (ER)/progesterone receptor (PR) subtype in 43 patients (53.1%), HER2 subtype in 18 patients (22.2%), and triple-negative breast cancer (TN) subtype in 20 patients (24.7%). Table 1 provides an overview of all patients in this study.

### 2.2. Representative Multicolored Immunostaining Images

As shown in Figure 1a, we used antibodies against CD68 (blue), CD163 (yellow), PD-L1 (orange), CD20 (pink), HLA (green), and pan-CK (red), as well as 4′,6-Diamidino-2-Phenylindole, Dihydrochloride (DAPI) to successfully stain tissue sections. We obtained the virtual image of a single staining to confirm that each staining was reasonably functional (Figure 1b–g).

### 2.3. Quantitative Comparison of Stromal Tumor-Associated Macrophages (sTAMs), Immune Modification, and B-Cell-Marker-Positive Cells among Breast Cancer Subtypes

We stained stromal areas near cancer cells for panmacrophage (CD68), M2 macrophage (CD163), immune modification (PD-L1), and pancytokeratin (CK) markers using a multicolored immunohistochemical method. The number of CD68-, CD163-, PD-L1-, and CD20-positive cells for the three breast cancer subtypes are shown in Appendix A. We next compared the numbers of CD68/CD163, CD68/PD-L1, and CD163/PD-L1 double-positive cells. We also successfully performed double immunostaining, as shown in Figure 2. In the ER/PR subtype, we found that CD68/CD163 double-positive cells were significantly increased compared with those in CD68/PD-L1 and CD163/PD-L1 double-positive cells (*p* < 0.001 and *p* < 0.001, respectively). We also detected that the number of CD68/PD-L1-positive cells was significantly higher than that of CD163/PD-L1-positive cells (*p* = 0.002). We observed a similar trend in the HER2 and TN subtypes (HER2: *p* < 0.001, *p* < 0.001, and *p* < 0.001; TN: *p* < 0.001, *p* < 0.001, and *p* = 0.002, respectively).

We classified and compared each marker among the three breast cancer subtypes. We found that CD68-positive cells were more abundant in the HER2 than in the ER/PR subtype (*p* = 0.047) (Figure 3a). In addition, CD163-positive cells were significantly more abundant in the TN than in the ER/PR and HER2 subtypes (*p* < 0.001 and *p* = 0.030, respectively), and significantly more abundant in the HER2 than in ER/PR subtype (*p* = 0.047) (Figure 3b). Furthermore, we noticed that PD-L1-positive cells were significantly more abundant in the TN compared with those in the HER2 subtype (*p* = 0.029) (Figure 3c), whereas no significant differences were observed regarding CD20-positive cells (Figure 3d). The proportion of each sTAM subpopulation among cancer subtypes is shown in Appendix A.

In addition, we quantified multiple combinations of cells positive for CD68, CD163, PD-L1, and CK in the stromal area, and compared the cell counts of sTAM subpopulations among the three cancer subtypes. We observed that CD68/CD163 double-positive cells had significantly higher cell numbers in the TN than in the ER/PR (*p* < 0.001) or HER2 (*p* = 0.048) subtypes (Figure 3e). Likewise, the TN subtype had significantly more CD163+/PD-L1+/CK- cells than the ER/PR subtype (*p* < 0.001) (Figure 3g). The proportion of each sTAM subpopulation among cancer subtypes is shown in Appendix A.

### 2.4. Quantitative Comparison of Intratumoral Tumor-Associated Macrophages (iTAMs), Immune Modification, and B-Cell-Marker-Positive Cells among Breast Cancer Subtypes

To analyze the intratumoral TME, we quantitatively compared iTAM cell counts using the CD68, CD163, PD-L1, CD20, and pan-CK markers in the three cancer subtypes. We observed that among the four cell markers, there was a trend towards fewer CD20-positive cells in all three subtypes (Appendix A). More specifically, we noticed that the ER/PR subtype had significantly more CD68/PD-L1 and CD163/PD-L1 than CD68/CD163 double-positive cells (*p* < 0.001, *p* = 0.010) (Appendix A). We observed a similar trend in the HER2 subtype (*p* = 0.002 and *p* = 0.029, respectively) (Appendix A). Likewise, in the TN subtype, CD163/PD-L1 double-positive cells were significantly more abundant than CD68/CD163 double-positive cells (*p* = 0.001) (Appendix A).

The number of cells expressing each marker among the three breast cancer subtypes is shown in Figure 4. In particular, we found that the TN subtype had significantly more CD163 cells than the ER/PR (*p* < 0.001) and HER2 (*p* = 0.021) subtypes, and the HER2 subtype had significantly more CD163 cells than the ER/PR (*p* = 0.048) subtype (Figure 4b). The number of CD20-positive cells was higher in the HER2 (*p* = 0.003) and TN (*p* < 0.001) subtypes compared with that in the ER/PR subtype (Figure 4d). The proportion of each iTAM subpopulation among cancer subtypes is shown in Appendix A.

In addition, we compared the number of cells with the combined expression of CD68, CD163, PD-L1, and pan-CK in the intratumoral area. We measured the number of iTAMs for each of the three subtypes and found that the HER2 and TN subtypes had higher numbers of CD68/CD163 double-positive iTAMs (*p* = 0.005 and *p* < 0.001, respectively) compared with those in the ER/PR subtype (Figure 4e). Similarly, the number of CD68+/PD-L1+/CK-/CD163- iTAMs was higher in the HER2 (*p* = 0.002) and TN (*p* = 0.004) subtypes relative to that in the ER/PR subtype (Figure 4f). The proportion of each iTAM subpopulation among cancer subtypes is shown in Appendix A.

### 2.5. Quantitative Comparison of Cancer Cells among Breast Cancer Subtypes

To further analyze the expression of PD-L1 and HLA in breast cancer cells, we combined the expression of pan-CK, PD-L1, and HLA in the three breast cancer subtypes and quantified the number of positive cells (Figure 5). We found that CK+/PD-L1+/HLA- cancer cells were more abundant in the TN than in the ER/PR subtype (*p* = 0.026) (Figure 5a). Furthermore, we detected that CK+/PD-L1+/HLA+ cancer cells were significantly more abundant in the TN than in the ER/PR and HER2 subtypes (*p* < 0.001 and *p* = 0.002, respectively) (Figure 5c). The proportion of each cancer cell subpopulation among cancer subtypes is shown in Appendix A.

### 2.6. Relationship between TAMs, Immune Modification, B-Cell-Marker-Positive Cells, and Recurrence in TN Subtype

We investigated the association between TAM cell counts and recurrence in the TN subtype, which is the subtype with the highest rate of recurrence and metastasis among the three cancer subtypes [32].

First, we investigated the association between single staining for each marker and recurrence-free survival rates (Appendix A). In this analysis, we did not detect any significant differences between high and low cell numbers. Interestingly, when we performed multiple staining analysis, we found that compared with patients with lower cell counts, those with higher numbers of CD68+/PD-L1+/CK-/CD163- iTAMs were correlated with an decreased recurrence rate (*p* = 0.048, Figure 6e).

## 3. Discussion

The biological properties of cancer cells depend on environmental factors surrounding the tumor, in addition to their genomic alterations, gene expression, and metabolism. Humoral factors around tumor cells (oxygen status, nutritional status, pH, cytokines, and hormones) and cellular factors such as immune cells and fibroblasts crosstalk with tumor cells to form the TME [33]. Because TAMs dynamically regulate both humoral and cellular responses, detailed profiling of TAMs, including functional analysis, can provide insights into the role of the TME in various cancers [33,34,35,36,37,38].

In this study, we quantified sTAMs and iTAMs using HTIA by classifying breast cancer into histological subtypes that affect prognosis and influence treatment strategies. In breast cancer, the presence of TAMs is more common in the HER2 and TN subtypes [39,40]. In this study, both sTAMs and iTAMs were also more prevalent in the HER2 and TN than in ER/PR subtypes, in consistency with the previous reports. Our quantitative evaluation of TAMs using multiple staining provided more precise evidence.

From a technical perspective, our study provided two breakthroughs that advanced the detailed analysis of TAMs. One was the evaluation of different sTAM and iTAM subpopulations, and the other was their quantitative evaluation by multiplex IHC (mIHC) using multiple markers. The former contributed to the spatial assessment within the tumor tissue, whereas the latter added depth to the biological function of TAMs. Regarding TILs, which together with TAMs are the main cellular factors of TME, a detailed definition of TILs was published by the International TILs Working Group in 2014. Interestingly, measurement of stromal TILs (sTILs) exhibited better reproducibility than that of intratumoral TILs (iTILs), and unlike iTILs, sTILs were not affected by tumor cell density or proliferation patterns; thus, sTILs were better evaluated [41]. However, recent studies have reported that both sTILs and iTILs are predictors of pathological response to neoadjuvant chemotherapy [42] and that CD8+ iTILs are poor prognostic factors [43]. This suggested that both sTILs and iTILs are independently involved in the formation of the TME and in determining clinical outcomes. In this study, pan-CK negative cells were extracted even if cancer cells expressed PD-L1. Therefore, this made possible the combination of multiple markers in the same section and the accurate quantification of both sTAMs and iTAMs. Our results showed that, with regard to CD68+/CD163+ double-positive sTAMs, their number was higher in the TN than in the HER2 and ER/PR subtypes, whereas the number of iTAMs was lower in the ER/PR than that in the TN and HER2 subtypes. Currently, it is not clear the means by which this difference affects TME; however, further studies on iTAMs in breast and other cancers are expected to clarify the effect of iTAMs and sTAMs on TME.

Furthermore, HTIA provided an objective quantitative analysis of the relationship between the expression of TAM markers and that of PD-L1. PD-L1 is a predictive marker of the therapeutic response to immune checkpoint inhibitors and is expressed not only in cancer but also in stromal cells, including TAMs. In breast cancer, the expression of PD-L1 is high in the TN subtype [44]; however, its prognostic significance has not been established [45,46,47]. Our analysis showed that the number of CD68+/PD-L1+/CK-/CD163- iTAMs was higher in the TN and HER2 compared with that in the ER/PR subtype (Figure 4f), with the TN subtype having a significantly lower risk of relapse (Figure 6e). Recently, the clinical significance of the expression of PD-L1 by TAMs and its impact on tumor immunity and immune checkpoint inhibitors has attracted much attention. PD-L1 has been reported to send constitutive negative signals to macrophages, resulting in an immunosuppressive cell phenotype [48]. Furthermore, PD-L1-expressing TAMs have been shown to be indicators of aggressive malignant potential in lung cancer [49,50]. In melanoma, the presence of CD163+/PD-L1+ TAMs led to a favorable ICI response [51]. More specifically, CD68+/PD-L1+/CK-/CD163- iTAMs were assumed to be PD-L1-expressing M1 macrophages that have antitumor functions, resulting in favorable clinical outcomes. Intriguingly, the number of CD68+/PD-L1+/CK-/CD163- sTAMs in the stromal area did not affect the recurrence rate, whereas that of CD163+/PD-L1+/CK- sTAMs, which are assumed to have potential as M2 macrophages, showed a trend toward poor prognosis, although the difference was not statistically significant. Therefore, it is possible that not only the expression of markers and specific phenotype of TAMs but also their localization within the TME might influence the clinical outcome, such as recurrence and prognosis, similar to TILs.

This study had several limitations. First, we examined TAMs at five random locations in a single specimen for all cases; hence, it is unclear whether the present results reflect the entire TME. Recent advances in image analysis technology have enabled the analysis of whole sections [52]. This technique is expected to enable a more accurate evaluation of the TME. Technically, staining background and non-specific bindings have not been sufficiently investigated in the multiple immunohistochemistry used in this study. It is also expected that high-throughput transcript analysis, Western blotting, or proteome analysis other than multiple immunostaining can be performed at the same time to obtain more evidence-based data. It also remains unclear which cytological features contribute to a favorable prognosis in CD68+/PD-L1+/CK-/CD163- TAMs that correlated with favorable clinical outcomes in this study. In addition, some patients with TN subtypes had received neoadjuvant chemotherapy, and the impact of this was not considered in this study. In addition, TILs present in the TME are known to affect tumor progression and treatment efficacy [7]. According to previous reports, a good prognosis is associated with a high density of CD8-positive T-cells in cancer nests in a variety of cancer types [53,54,55,56]. Additionally, in various human malignancies, including breast tumors, the presence of TILs has been found to be a predictor of the response to neoadjuvant chemotherapy [57,58]. HTIAs for both TILs and TAMs will help elucidate the complexity of the TME in breast cancer.

## 4. Materials and Methods

### 4.1. Samples

The specimens used in this study were formalin-fixed paraffin-embedded (FFPE) specimens from 81 patients with invasive ductal breast carcinoma diagnosed and surgically treated at Tokyo Medical and Dental University Hospital between 2014 and 2017. These specimens were fixed in 10% neutralized formalin and used for histopathological examination. Residual specimens were used as study samples. Briefly, FFPE tissues were cut to 4 µm thick and placed on silane-coated glass slides. Informed consent was obtained from all patients. The study was approved by the Ethics Committee of Tokyo Medical and Dental University (approval No, M2018-141) and the Ethics Committee of Daiichi Sankyo RD Novare Co., Ltd. (1-16-13 Kitakasai, Edogawa-ku, Tokyo 134-0081, Japan) (approval No, N18-0082-00). All study designs were conducted under the regulations of the committees.

### 4.2. Multiplex Immunohistochemistry

An automated slide staining machine BOND RX (Leica Biosystems, Nußloch, BW, Germany) and an Opal 7-color Automation IHC kit (Perkin Elmer, Waltham, MA, USA) were used for mIHC staining. The following primary antibodies were used: CD68 (1:2000, clone KP-1, Agilent, Tokyo, Japan), CD163 (1:100, clone 10D6, Leica Biosystems), PD-L1 (1:100, clone SP142, abcam, Cambridge, MA, USA), CD20 (1:1, clone L26, Agilent, Tokyo, Japan), HLA class I (1:800, clone EMR8-5, abcam, Cambridge, MA, USA), and pan-cytokeratin (CK) (1:4, clone AE1/AE3, Agilent, Tokyo, Japan). Staining was performed using the Opal 7-color automation IHC kit and BOND research detection kit (Leica Biosystems Nußloch, BW, Germany), according to the manufacturer’s instructions. Primary antibodies were incubated for 30 min at 25 °C. Slides were mounted using a ProLong Diamond Antifade Mountant (Thermo Fisher Scientific, Waltham, MA, USA).

### 4.3. Quantitative Image Analysis

Multiplex staining was performed on 81 specimens, followed by multispectral image analysis using the Vectra Polaris automated quantitative pathology imaging system (PerkinElmer, Waltham, MA, USA). Whole-slide images of each slide were captured at 10× magnification to select high-magnification multispectral images (MSI) at 20× (0.5 μm/pixel resolution, 931 μm × 698 μm) using the Phenochart viewer (version 1.0, PerkinElmer, Waltham, MA, USA). For quantitative image analysis, the top 5 TAM hotspot fields were selected as regions of interest (ROIs) per slide for each section and scanned for MSI. None of the sections contained non-neoplastic epithelial cell components. For the 5 scanned areas, image files were created using Vectra Polaris and analyzed using the image analysis software inForm (version 2.4.0., Perkin Elmer, Waltham, MA, USA). Details of these procedures have been previously described [27]. Then, scoring and phenotyping methods were performed to count the number of positive cells with a signal over a particular threshold for each marker. In the scoring method, positive/negative thresholds were visually set based on fluorescence intensity values, and the number of cells that were single-positive and double-positive, respectively, were counted using the image analysis software inForm. In selected ROIs, the density of single-positive (CD68, CD163, PD-L1, CD20, tumor area HLA, and pan-CK) and double-positive (CD68-CD163, CD68-PD-L1, CD163-PD-L1) cells were calculated in both stromal and intratumoral regions. In the phenotyping method, a combination of multiple marker expressions was used to identify cells of a particular phenotype on the image, several cells were manually selected for each phenotype of interest, and cells of each phenotype were counted using an inForm-based machine learning algorithm. In this study, cells in stromal and intratumoral regions were classified into the following phenotypes: macrophages (CD68+/PD-L1+/CK-/CD163 cells and CD163+/PD-L1+/CK- cells) and tumor cells (CK+/PD-L1+/HLA- cells, CK+/PD-L1-/HLA+ cells, and CK+/PD-L1+/HLA+ cells).

### 4.4. Data Analysis and Statistical Procedures

We analyzed all scoring data generated by inForm. The number of TAM marker-positive cells per mm^2^ in each cancer subtype was compared. One-way ANOVA and Tukey’s multiple comparison test were used to determine the statistical significance of the unpaired data in the scoring of each marker. Kaplan–Meier analysis was performed by classifying each marker as high (*n* = 10) or low (*n* = 10) except for the CK+/PD-L1+/HLA- subpopulation in cancer cells, which was classified as more than one cell (*n* = 5) or none (*n* = 15). Statistical analyses were performed using GraphPad Prism version 9.2 (GraphPad Software, San Diego, CA, USA); *p* < 0.05 was defined as statistical significance.

## 5. Conclusions

Taken together, our findings provided objective basic data on the TME in breast cancer, especially the status of TAMs, and indicated that subsets with specific localization and markers might be used as predictors of recurrence risk in cancer subtypes with poor prognosis. Future studies will focus on establishing a more detailed TAM classification-based subset in cancer subtypes to further elucidate the unknown functions of the TME. In addition, a prospective cohort with a large number of patients is expected to provide evidence for the validity of the use of the TME as a prognostic factor and for predicting the effect of ICIs on cancer treatment.

## Figures and Tables

**Figure 1 ijms-23-13235-f001:**
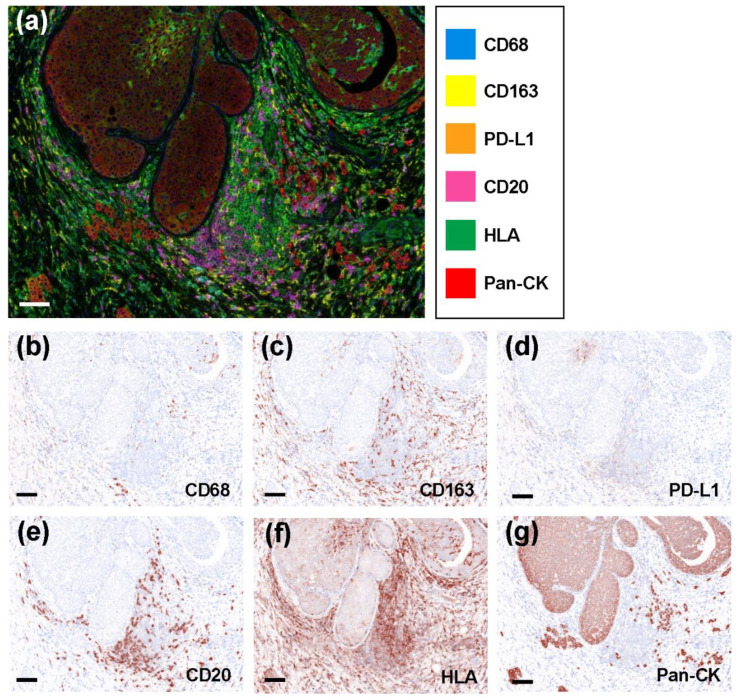
Representative multicolored immunostaining and virtual pathological images. (**a**) Composite image (7-color image) of representative lesion. Virtual pathological image of (**b**) CD68, (**c**) CD163, (**d**) PD-L1, (**e**) CD20, (**f**) HLA, and (**g**) pan-CK staining in a single tissue section. Brownish-colored areas are positive signal. Scale bars indicate 100 μm.

**Figure 2 ijms-23-13235-f002:**
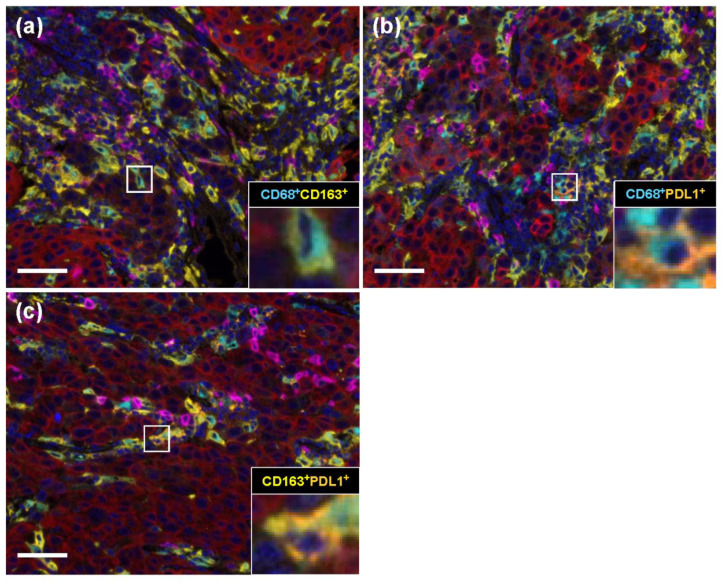
Representative TAMs and immune modification marker double-positive cells obtained using automated quantitative pathology imaging system and image analysis software. Representative images of immunostained double-positive cells: (**a**) CD68 (blue)/CD163 (yellow), (**b**) CD68 (blue)/ PD-L1 (orange), and (**c**) CD163 (yellow)/PD-L1 (orange) double-positive cells in breast cancer tissues. Scale bars indicate 100 μm.

**Figure 3 ijms-23-13235-f003:**
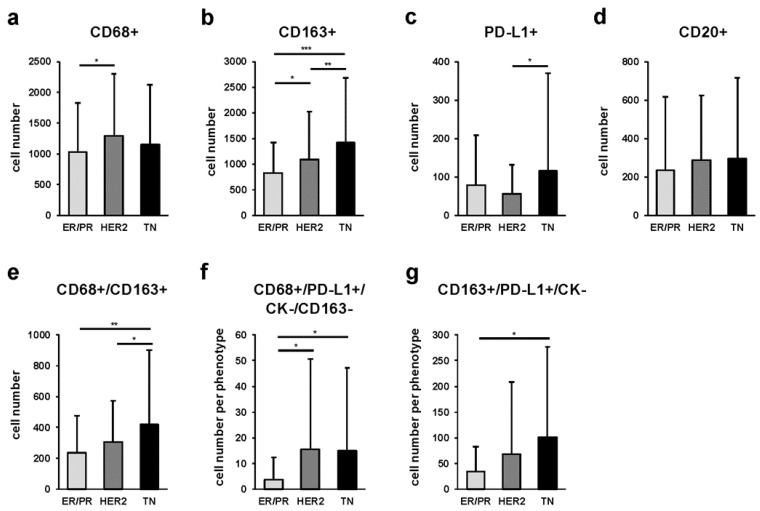
Comparison of single- and multiple-stained stromal TAMs (sTAMs), immune modification, and B-cell-marker-positive cells among ER/PR, HER2, and TN subtypes. Average number of single-marker-positive cells: (**a**) CD68+ (* indicates *p* = 0.047), (**b**) CD163+ (*, ** and, *** indicate *p* = 0.047, *p* = 0.030, and *p* < 0.001, respectively), (**c**) PD-L1+ (* indicates *p* = 0.029), and (**d**) CD20+ cells. Average number of multiple-marker-positive cells: (**e**) CD68/CD163 double-positive (* and ** indicate *p* = 0.048 and *p* < 0.001, respectively), (**f**) CD68+/PD-L1+/CK-/CD163- (* indicates *p* < 0.001), and (**g**) CD163+/PD-L1+/CK- (* indicates *p* < 0.001) per 1 mm^2^ at 5 locations in each subtype; *p*-value was calculated using one-way ANOVA test and Tukey’s multiple comparison test. Error bars represent standard deviation.

**Figure 4 ijms-23-13235-f004:**
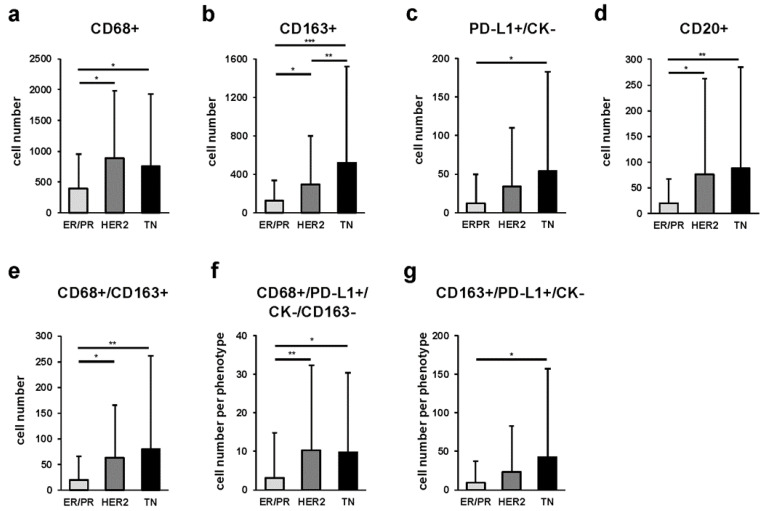
Comparison of single- and multiple-stained intratumoral TAMs (iTAMs), immune modification, and B-cell-marker-positive cells among ER/PR, HER2, and TN subtypes. Average number of single-marker-positive cells: (**a**) CD68+ (* indicates *p* < 0.001), (**b**) CD163+ (*, ** and, *** indicate *p* = 0.048, *p* = 0.021, and *p* = < 0.001, respectively), (**c**) PD-L1+ (* indicates *p* < 0.001), and (**d**) CD20+ (* and ** indicates *p* = 0.003 and *p* < 0.001, respectively) cells. Average number of multiple-marker-positive cells: (**e**) CD68/CD163 double-positive (* and ** indicate *p* = 0.005 and *p* < 0.001, respectively), (**f**) CD68+/PD-L1+/CK-/CD163- (* and ** indicate *p* = 0.004 and *p* = 0.002, respectively), and (**g**) CD163+/PD-L1+/CK- (* indicates *p* < 0.001) iTAMs per 1 mm^2^ at 5 locations in each subtype; *p*-value was calculated using one-way ANOVA test and Tukey’s multiple comparison test. Error bars represent standard deviation.

**Figure 5 ijms-23-13235-f005:**
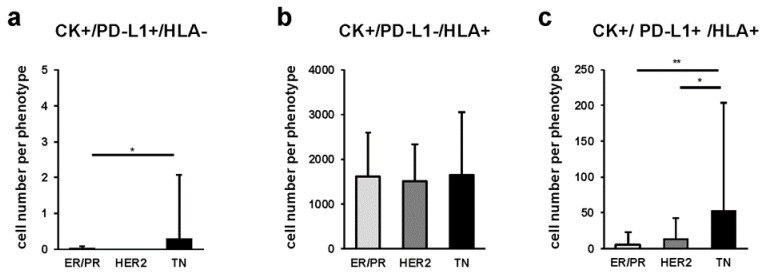
Comparison of multiple-marker-stained cancer cells among ER/PR, HER2, and TN subtypes. Average number of cells stained for multiple markers: (**a**) CK+/PD-L1+/HLA- (* indicates *p* = 0.026), (**b**) CK+/PD-L1-/HLA+, and (**c**) CK+/PD-L1+/HLA+ (* and ** indicate *p* = 0.002 and *p* < 0.001, respectively) per 1 mm^2^ at 5 locations in each subtype; *p*-value was calculated using one-way ANOVA test and Tukey’s multiple comparison test. Error bars represent standard deviation.

**Figure 6 ijms-23-13235-f006:**
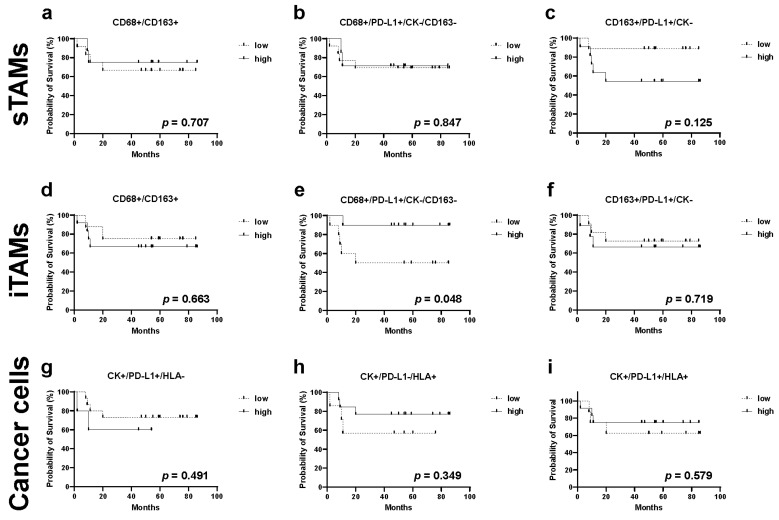
Kaplan–Meier analysis for recurrence-free survival categorized by the number of TAMs (sTAMs and iTAMs), immune modification, and B-cell and cancer cell markers in triple-negative cases. Each subpopulation combining markers of TAMs, B-cells, and cancer cells was classified into “low” or “high”: (**a**) CD68+/CD163+ sTAMs low, <280 cells; high, ≥280 cells; (**b**) CD68+/PD-L1+/CK-/CD163- sTAMs low, <3 cells; high, ≥3 cells; (**c**) CD163+/PD-L1+/CK- sTAMs low, <50 cells; high, ≥50 cells; (**d**) CD68+/CD163+ iTAMs low, <25 cells; high, ≥25 cells; (**e**) CD68+/PD-L1+/CK-/CD163- iTAMs low, <1 cells; high, ≥1 cells (*p* = 0.048); (**f**) CD163+/PD-L1+/CK- iTAMs low, <10 cells; high, ≥10 cells; (**g**) CK+/PD-L1+/HLA- tumor cells low, 0 cell; high, ≥1 cell; (**h**) CK+/PD-L1-/HLA+ tumor cells low, <1500 cells; high, ≥1500 cells; and (**i**) CK+/PD-L1+/HLA+ tumor cells low, <12 cells; high, ≥12 cells.

**Table 1 ijms-23-13235-t001:** Clinicopathological features in this study.

Clinicopathological Features		ER/PR (*n* = 43)	HER2 (*n* = 18)	TN (*n* = 20)
Age	Median (Range)	59 (35–86)	54 (33–71)	63 (36–88)
pT classification	T1	22	11	8
	T2	16	7	8
	T3	4	0	3
	T4	1	0	1
Neoadjuvant chemotherapy	+	6	2	7
	−	37	16	13
Lymph node metastasis	Positive	13	6	4
	Negative	28	10	12
	NX	2	2	4

## Data Availability

Not applicable.

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
