# Peer review of "Spatial and Quantitative Analysis of Tumor-Associated Macrophages: Intratumoral CD163-/PD-L1+ TAMs as a Marker of Favorable Clinical Outcomes in Triple-Negative Breast Cancer"

_ijms, 2022, doi:10.3390/ijms232113235_

Round 1

Reviewer 1 Report

The paper is overall in a acceptable condition. Please address the follwing comments.

1. All images need scale bar. 

2. How exactly do the authors quantified the staining images? Please include this information in section 2.

3. Please add discussion on staining background and non-specific bindings.

Author Response

Response to Reviewer 1

  1. All images need scale bar. 

>Scale bars were added in figure 1a-g and figure 2a-c. Also, we added a note about the scale bar in the figure legend column.

  1. How exactly do the authors quantified the staining images? Please include this information in section 2.

>The detail of quantification method of the cells used in this study was added in Section 2.

  1. Please add discussion on staining background and non-specific bindings.

>In the Discussion section, we mentioned staining background and non-specific bindings as a limitation.

Reviewer 2 Report

The article "Spatial and quantitative analysis of tumor-associated macro-2 phages: Intratumoral CD163-/PD-L1+ TAMs as a marker of favorable clinical outcomes in triple-negative breast cancer" is well presented. However, the graphs describe the number of cells expressing each marker among the 3 breast cancer subtypes. What is the control in those panels of data? 

I suggest also, inserting the legends in the supplement data.

It could be also useful as a double control, adding other experimental analyses among the 3 breast cancer subtypes, (PCR or immunoblot )at least of a set of markers analyzed. 

Author Response

Response to Reviewer 2

  1. The article "Spatial and quantitative analysis of tumor-associated macro-2 phages: Intratumoral CD163-/PD-L1+ TAMs as a marker of favorable clinical outcomes in triple-negative breast cancer" is well presented. However, the graphs describe the number of cells expressing each marker among the 3 breast cancer subtypes. What is the control in those panels of data? 

>In this study, the cell counts were quantified using two methods: "scoring" and "phenotyping. Both of these methods are well established in multiple immunohistochemistry. "Scoring" and "phenotyping" have been added in the Materials and methods section.

  1. I suggest also, inserting the legends in the supplement data.

>Supplementary figure and table were added at the end of the main manuscript.

  1. It could be also useful as a double control, adding other experimental analyses among the 3 breast cancer subtypes, (PCR or immunoblot ) at least of a set of markers analyzed. 

> Thank you very much for your valuable comments. As the reviewer suggested, we expect to obtain more evidenced-based data by using high throughput transcript analysis, western blotting, or proteome analysis, as well as multiple immunostaining. We have added this point to the limitation section of the discussion.